# Supportive Care Needs of Patients with Breast Cancer Who Self-Identify as Black: An Integrative Review

**DOI:** 10.3390/curroncol32100580

**Published:** 2025-10-18

**Authors:** Etienne Oshinowo, Emily Peterson, Michelle Audoin, Jennifer Ryan, June Buckle, Clare Cruickshank, Jennifer Jones, Lisa Malinowski Kamran, Aisha Lofters, Patricia Russell, Leila Springer, Danielle VandeZande, Ashanté Lakey, Laura Burnett, Melanie Powis

**Affiliations:** 1Cancer Quality Lab (CQuaL), Princess Margaret Cancer Centre, Toronto, ON M5G 1X6, Canada; 2Patient and Community Partner, Canada; 3Division of Medical Oncology and Hematology, Princess Margaret Cancer Centre, Toronto, ON M5G 1X6, Canada; 4The Olive Branch of Hope, Toronto, ON M3K 1B9, Canada; 5Cancer Rehabilitation and Survivorship Program, Princess Margaret Cancer Centre, Toronto, ON M5G 2M9, Canada; 6Gilda’s Club Greater Toronto, Toronto, ON M6P 1Y0, Canada; 7Women’s College Hospital, Toronto, ON M5S 1B2, Canada; 8Department of Family & Community Medicine, University of Toronto, Toronto, ON M5G 1V7, Canada; 9Canadian Cancer Society, Toronto, ON M4V 1Y7, Canada; 10Institute of Health Policy, Management and Evaluation, University of Toronto, Toronto, ON M5T 3M6, Canada

**Keywords:** breast cancer, black, supportive care, informational resources, nominal group, needs assessment

## Abstract

**Simple Summary:**

Black women are more likely to be diagnosed with breast cancer at a younger age and with more aggressive cancers. Once diagnosed, they are less likely to receive appropriate treatments for their cancer and are more likely to experience dose reductions, dose holds, and early treatment termination, which can negatively impact survival. However, little is currently known about the specific supportive care and informational needs of this population, particularly in the Canadian healthcare system context. To address this gap, we performed an integrative review to synthesize the findings of prior needs assessments of patients with breast cancer who were reported to be Black from the published literature and integrated them with the findings of a pan-Canadian needs assessment and prioritization exercise conducted by our team. This work identified patient navigation, financial support, access to culturally tailored information and education, culturally relevant care, racialized data, and emotional support as high-priority needs.

**Abstract:**

Black-identifying patients face many barriers to the receipt of equitable breast cancer care; however, little is currently known about the unique needs of this patient population, particularly in Canada. To address this gap, we identified and thematically grouped constructs from the published literature reporting on the needs of Black-identifying patients with breast cancer and compared these findings to a list generated through a virtual nominal consensus group (NG) attended by Canadian patients with breast cancer who self-identified as Black (n = 3). A scoping review was undertaken, and relevant citations published from database inception until January 2025 were identified from MEDLINE, Embase, and CINAHL. The literature review yielded 34 articles from the United States and identified 15 constructs consistent with the NG, which spanned the cancer continuum from screening to survivorship. The NG identified four additional constructs that were not found in the literature: advocacy and outreach, communication and health literacy, comorbidities and personalized care, and end-of-life care. The final set of constructs was then validated and prioritized by an expert panel consisting of patients with lived experience and relevant community partner organizations (n = 9) to drive future research, advocacy, and policy work. Patient navigation was identified as the top need, with financial support, access to culturally tailored information and resources, culturally relevant care, racialized data for treatment decision-making, and emotional support identified as high-priority needs.

## 1. Introduction

Breast cancer remains one of the most prevalent malignancies amongst women [1], constituting approximately 25% of incident cancer cases [2] and 14% of cancer-related deaths [3]. Black women are disproportionately affected by aggressive sub-types and are more likely to be diagnosed at a younger age [4] and a higher stage [5]. Once diagnosed, many structural barriers prevent Black-identifying patients from receiving equitable breast cancer care [6]. Fewer Black women receive appropriate genetic testing [7], timely systemic therapy [8], treatment with targeted drugs [9], or access to interventional trials [10]. Additionally, there is growing evidence that Black women experience a higher incidence of serious treatment-related toxicity, such as anthracycline-induced cardiotoxicity [11,12,13], which, with sub-optimal management, can lead to higher rates of dose reductions, dose holds, and early termination, negatively impacting both treatment efficacy and survival [14,15]; however, a lack of representation in prospective trials [16], a lack of race and ethnicity data in routinely collected administrative health data [17], and a lack of targeted research [18] limit current knowledge. In addition to clinical differences and structural barriers to care, social determinants of health, including anti-Black racism, economic insecurity, and medical mistrust, further exacerbate disparities in the health outcomes of Black women and necessitate additional supportive resources [19].

Addressing these multi-layered barriers requires advancements in current clinical research that are inclusive of underrepresented populations [20], as well as systemic interventions that prioritize representation, equity, and culturally tailored models of cancer and supportive care [21]. However, much of the literature to date has focused on describing care disparities rather than addressing underlying causes or centering research on tailored solutions to reduce inequities. As such, little remains known about the unique supportive care and informational needs of this patient population, as there has been no published evidence synthesis to date, particularly in the Canadian single-payer, publicly funded healthcare system context [22].

To address this gap, we undertook an integrative review [23]. We conducted a scoping review to identify and synthesize what is known about the needs of Black patients with breast cancer in the published literature. Findings were then grouped thematically across the stages of the cancer continuum from screening and diagnosis to survivorship care. Since all of the retained studies were completed in the United States, we conducted a virtual nominal consensus group to identify the supportive care and informational needs of patients with breast cancer who self-identify as Black from across Canada and compared these findings to constructs identified in the literature [24]. The final set of needs was validated and prioritized through a series of workshops with a panel of patients with lived experience and community partner organizations to direct future research, advocacy, and policy efforts.

## 2. Materials and Methods

This study leveraged integrative review research methods [23,25] and included a scoping review of the literature, coupled with nominal consensus groups, and a validation and prioritization exercise through a series of workshops with patient partners with lived experience and relevant community advocacy organizations. The literature review did not require ethics approval; however, the nominal consensus group was approved by the University Health Network Research Ethics Board (UHN REB#: 24-5472), and the validation and prioritization workshops were approved through a separate application to the University of Toronto Research Ethics Board (Protocol#: 48281).

To identify existing studies reporting on the needs of Black-identifying patients with breast cancer, we undertook a scoping literature review. Relevant citations published from database inception to January 2025 were retrieved from MEDLINE, Embase, and CINAHL using search terms grouped by key concept (breast cancer, support needs, and Black), with syntax and headings translated as appropriate across databases (Appendix A). Resultant citations were imported into Covidence (Veritas Health Innovation; Melbourne, Australia) for review, and duplicates were removed. The review was carried out in accordance with the PRISMA extension guidelines for scoping reviews [26]. Studies were included if they reported on primary research examining the needs of Black-identifying patients with breast cancer of any gender or sex. The search was restricted to articles published in English and reporting on adult populations. Articles were excluded if they were literature reviews, protocols for needs assessments without accompanying findings, commentaries, letters to the editor, or abstracts only. Additionally, articles reporting on population-level cohort studies that derived patient needs based on observed gaps in care, rather than soliciting needs directly from patients, were excluded.

Titles, abstracts, and full texts were screened for inclusion by two reviewers (EP and MP); no conflicts in inclusions between reviewers were identified. Details of retained articles were abstracted using a standardized abstraction form into Excel (Microsoft, Redmond, DC, USA) by one reviewer (EO). A second abstractor (MP) extracted data from a random sub-set of 10 articles to evaluate the reliability of the abstraction process; there were no discrepancies identified between reviewers. Findings were inductively coded, then thematically grouped into constructs and concepts and mapped to the stages of the breast cancer continuum: (1) screening and diagnosis; (2) active treatment; (3) survivorship; (4) end-of-life; or (5) overarching.

In parallel, we undertook a nominal consensus group [24] to identify the needs of patients with breast cancer who self-identified as Black from across Canada. Potential participants were recruited through an email invitation disseminated to members of the Patient and Caregiver Partners listserv of the Canadian Cancer Society. To be eligible to participate, participants had to be ≥18 years of age, have a prior diagnosis of breast cancer for which they had received treatment in the previous 12 months, self-identify as Black, reside in Canada, and be proficient in spoken English. The nominal group was carried out virtually on Zoom for Healthcare (Zoom Communications, San Jose, CA, USA). The group was facilitated by members of the study team trained in nominal group technique (MA, JR, and MP). At the start of the session, participants were provided with a prompt: “*If you received any supportive resources or services during your treatment with chemotherapy what services did you find helpful? If you did not receive any supports, what support resources or services do you think would be/have been helpful to you during and after chemotherapy?*” Following silent ideation, participants underwent iterative round-robin-style rounds of ideation until all ideas were recorded, then serially discussed each idea [27,28,29]. Ideas were reorganized and combined or parsed by the participants as they reached consensus on the final set of needs. The session was recorded and transcribed verbatim; supportive quotes for each of the needs identified in the session were extracted from the transcript. The final set of needs identified by the nominal consensus group was then compared against the findings of the literature review to identify areas of concordance and discordance [23,25].

The final set of needs identified by the nominal consensus group was then presented to a panel consisting of patients with lived experience and relevant community partner organizations, assembled to leverage a community-based participatory research approach [30] to co-design a supportive care program for patients with breast cancer who self-identify as Black. Through a series of workshops, panel members validated the list of supportive care and informational needs and undertook consensus discussions to arrive at a final ranking of the needs in order of priority from highest to lowest.

## 3. Results

### 3.1. Details of Retained Articles

The scoping literature search identified 7508 articles; after duplicate removal and review, 34 met inclusion criteria and were retained for analysis (Figure 1). All retained studies were conducted in the United States and focused on Black patient populations residing in specific regions, either the American South (42.9%; 15/34) or on the East Coast (29.4%; 10/34; Table 1). Sample sizes were generally small, with 47% (16/34) of articles reporting on samples of ≤25 patients (range: 9–1056). None of the articles identified needs across all stages of the cancer continuum or included all of the identified constructs. Details of all the retained articles are included in Appendix A. The majority of articles identified supportive care and interventional needs relevant to all stages of the cancer continuum (overarching; 94.1%; 32/34), screening and diagnosis (76.5%; 26/34), or active treatment (70.6%; 24/34). Few articles included needs from the survivorship phase (23.5%; 8/34), while none of the articles specifically discussed end-of-life care or palliation.

### 3.2. Supportive Care and Informational Needs Common to the Literature and Nominal Consensus Group (NG)

The literature review identified 15 unique informational and supportive care needs, many of which spanned the breast cancer continuum. Findings of the literature review are summarized in Figure 2 and below.

#### 3.2.1. Overarching Needs

The literature review findings highlighted the need for emotional support (79.4%; 27/34) across the breast cancer continuum, which included access to formal psychosocial supports (38.2%; 13/34) who can be confided in and who will actively listen, to address changing relationships, social needs, and mental health across the cancer journey (Table 2). In line with the literature, NG participants stressed the need for psychosocial supports to be provided by clinicians who are knowledgeable of and reflect the Black community. NG participants also identified the need to provide upfront information for patients on the types of supports that are available, how long the wait for psychosocial support is likely to be, and the frequency and duration of support that will be provided. The literature review identified the need for more targeted emotional support resources specifically for children and families (35.3%; 12/34) and caregivers (2.9%; 1/34). NG participants highlighted the impact that cancer care can have on existing family dynamics and stressed the need for tailored family support groups, as well as the availability of childcare to reduce barriers to accessing cancer treatment and supportive resources. The literature underscored the need for access to peer support (61.8%; 21/34) for practical support and advice from others with lived experience with cancer who had faced similar challenges. NG participants reflected that it was helpful to connect with community support groups and peers who had lived similar experiences to themselves and understood what they were going through. However, they highlighted the need for better engagement with, and integration within, the Black communities and Black-focused community health organizations to ensure patients have a safe and welcoming space.

The findings emphasized a desire for spiritual support (47.1%; 16/34) to help describe a sense of meaning in the cancer experience in line with the patients’ belief system; however, this need was much less pronounced in the NG than in the literature, which may reflect both the representativeness of the NG sample and the greater diversity of religious beliefs amongst patients from Canada. To reduce barriers to accessing care, reliable transportation was needed to get to and from treatments and supports (8.8%; 3/34), and financial resource information and supports (26.5%; 9/34) for out-of-pocket cancer and treatment-related costs. NG participants identified information and resources to support financial planning and critical injury insurance, and financial assistance as needed financial supports. In the literature, patients emphasized the need for culturally relevant and trauma-informed approaches to care (14.7%; 5/34), as well as improved quality of patient–physician communication and relationships (35.3%; 12/34) through shared decision-making and culturally sensitive communication approaches. NG participants stressed that seeing any gains in breast cancer care equity would require healthcare system adoption of anti-racist and anti-oppressive reflexivity and approaches as a precursor.

#### 3.2.2. Screening and Diagnosis

Specific to the screening and diagnosis phases, patients identified the need for culturally and ethnically tailored information and education on breast screening and cancer risk, treatment efficacy, and treatment options (67.6%; 23/34). Participants in the NG noted that most easily identifiable sources of information are from the United States and that access to information can help drive self-advocacy efforts. The NG highlighted similar informational needs as in the literature but also specifically called out the need for information on what to expect during treatment, how to interpret test results, information on types and rates of treatment-related toxicities amongst Black patients, scarring and skin toxicities on darker skin tones, prosthetics, genetic testing, end-of-life care and hospice, and resources for Black hair, including racially tailored information on the effectiveness of cold capping.

Patients in the literature and NG were interested in access to racialized data on treatment-related toxicities, risks, and clinical outcomes to inform treatment decision-making (11.8%; 4/34). Additionally, both the literature review (8.8%; 3/34) and NG highlighted the need for information and resources to support fertility preservation. In the NG, patients highlighted the need for sufficient time to think about and make informed decisions, and the pivotal roles that emotional and financial support play in fertility preservation discussions and choices. Participants in the NG highlighted the cultural significance of family and children and the potential for loss of identity and community alongside loss of fertility post-treatment. The NGs identified two additional constructs that were not included in the literature review: communication and health literacy, and advocacy and outreach, which are likely reflective of sociodemographic and contextual differences between our NG sample and samples from previously published articles from the United States, which were primarily carried out in the South and/or in jurisdictions with lower median incomes and literacy rates, and in which proportions of Black-identifying healthcare providers are higher. Participants highlighted the need for better patient–clinician communication and the role that health literacy plays in patients’ abilities to self-advocate or find relevant and trustworthy information. They highlighted the pivotal role that advocacy and community outreach play in improving rates of screening and diagnosis in the Black community and stressed the need for the cancer system to make more intentional efforts to engage with the Black community.

#### 3.2.3. Active Treatment

Patient navigators were identified by both the literature (38.2%; 13/34) and the NG as needs, particularly during treatment decision-making and active treatment. NG participants conceptualized the role of the navigator as someone to walk through treatments, tests, and results with patients; connect them to tailored resources; help them to navigate the healthcare system, particularly between providers and institutions; use lay terms and answers questions; and act as a bridge between the cancer system and the community. Participants stressed how exhausting self-advocacy can be and identified that navigators could alleviate some of that burden by attending appointments with patients and advocating on their behalf. Fitness and nutrition were identified as needs in both the literature (26.5%; 9/34) and in the NG. NG participants identified the need for information on best practices for fitness before, during, and after treatment, how to build stamina to stay on treatment, access to physiotherapy, and the need for culturally tailored information about what to eat when experiencing treatment-related toxicities.

Body image was identified by the literature (20.6%; 7/34); NG participants discussed the relationship between body image, hair texture, and identity. Participants discussed the cultural significance of hair and beauty and the need for emotional support to work through the grief of loss of body parts and body image. Career and employment needs (literature: 2.9%; 1/34) identified in the NG included resources and advice on patients’ rights when navigating critical illness with employers and on how to manage working while experiencing treatment-related toxicities. Sexual health information and education on intimacy and post-treatment reproductive health were identified as needs in the literature (17.6%; 6/34) but were not identified in the NG, which may be reflective of the NG sample size. On the other hand, NG participants identified the need for whole-person, personalized care that considers their comorbid conditions.

#### 3.2.4. Survivorship

Both the literature (23.5%; 8/34) and NG identified the need for improved supports during the transition from active treatment to survivorship. NG participants discussed the fear of cancer recurrence, the sense of uncertainty about what happens or should happen once treatment ends, the evaporation of formal, hospital-based supports at the end of treatment, and the positive role that warm hand-offs to peer support and community-partner organizations could play in improving care.

#### 3.2.5. End of Life

The literature did not identify any constructs specific to end-of-life care. NG participants discussed the need for information on accessing palliative care, hospice care, and medically assisted dying. Participants discussed the lack of information on what supports are available and the associated costs, as well as the need for culturally competent approaches to discussing end-of-life care.

### 3.3. Prioritization of Supportive Care and Informational Needs

Findings of the literature review and NG were validated through a series of workshops with a panel consisting of patients with lived experience and relevant community partner organizations (n = 9). Through discussions, the identified needs were ranked and re-ranked in order from highest to lowest priority needs until consensus on priority was reached. The panel identified patient navigation as the foundational need, followed, in order from highest priority, by (1) financial support; (2) culturally tailored education and information; (3) culturally relevant care; (4) access to racialized data; and (5) psychosocial support (Figure 3).

## 4. Discussion

Numerous, persistent disparities in the receipt of equitable breast cancer care for Black patients [4,5,6,7,8,9,10] highlight the ongoing need for tailored interventions aimed at overcoming the structural and sociodemographic barriers to care. However, there has been little published to date on the unique needs of this patient population, and all of the studies so far have been undertaken in the United States. As such, needs may not translate well to the Canadian healthcare system, given the differences inherent in the universal healthcare system context [43], the differing role of private insurance in cancer and supportive care coverage [44], the comprehensiveness of available social supports, breast cancer clinical outcomes [45], and population diversity [46]. In addition to the 15 needs in common between the literature review and NG, we identified four additional needs in the NG that were not identified in the literature: advocacy and outreach, communication and health literacy, comorbidities and personalized care, and end-of-life care. These needs reflect broader structural and cultural barriers that affect Black patients. Through a prioritization exercise, patient navigation was seen as a foundational need, with financial support, tailored education and information, culturally relevant care, access to racialized data, and emotional support as the highest-ranking needs. To our knowledge, our study is the first to synthesize the literature around supportive care needs of Black-identifying patients with breast cancer, and the first published needs assessment of this population in the Canadian context. The results of our study demonstrate a critical gap between the documented needs of predominantly United States-based literature and the lived experiences of Canadian women. Understanding the priorities of Black-identifying patients with breast cancer is an instrumental step in developing community-centered interventions to address these needs [47], alongside policy changes and funding opportunities that support their implementation and evaluation and help to break down structural barriers to care.

The needs for navigation, culturally tailored information and education, and culturally relevant care were explicitly called out as community priorities through our consensus discussions. Navigation fatigue has been described as a growing issue prevalent in marginalized patients. Repeated interactions with fragmented healthcare support create exhaustion and disengagement [48]. For Black-identifying patients, these prolonged negative interactions with the system are compounded by experiences of racism with care providers and the burden of having to educate providers on their cultural needs [48]. Our findings align with both the Canadian Cancer Society’s Breast Cancer Survey and Focus Group Data [49] and the Uncovered: A Breast Recognition Project [50], which specifically identifies the need for patient navigation to support patients who self-identify as Black, to improve access to and connect them with culturally relevant information and treatments and culturally appropriate supportive care services. There is substantial evidence of the positive impacts of Patient Navigators on improving access to and receipt of quality care for Black patients undergoing breast cancer treatment in other jurisdictions [51,52]. Navigation programs have been shown to shorten the time to treatment initiation, improve adherence, enhance patient satisfaction and quality of life, and decrease unscheduled acute-care utilization, particularly among racialized groups [52,53]. Findings from our NG stressed that proper navigation would require culturally embedded approaches that take into account the structural inequities Black women face. This would require training navigators in anti-racist and trauma-informed practices as well as connecting women to community and other referral networks [54]. A study by Pal et al. [55] demonstrated that scalable, culturally relevant educational interventions, such as web-based tools, can empower Black breast cancer patients with the resources and knowledge required to make informed decisions about their care. Another project, which supports women across the United States and is known as the Susan G. Komen Foundation’s Black Women’s Health Equity, which echoes the need for navigators to be culturally informed. The foundation notes that without addressing financial toxicity, systemic mistrust, lack of support networks, and cultural disconnection, navigation programs will ultimately be ineffective for Black patients [56].

Financial impacts of cancer on patients are substantial, with the Canadian Cancer Society estimating the lifetime out-of-pocket costs for a patient at approximately 33,000 CAD [57]. The need for improved financial supports for patients with breast cancer was reflected in both the literature and NG findings. Financial toxicity disproportionately affects Black patients, and ultimately, financial strain can influence treatment choices, adherence, and outcomes. This disparity is linked to loss of income post-diagnosis and even persists after controlling for clinical and socioeconomic factors [58]. Financial navigation interventions that provide support for insurance, financial counselling, and accommodations for employment can mitigate financial burden and substantially reduce financial toxicity [59].

The need for improved emotional supports for patients with breast cancer was identified in both the literature review and NG, prioritized by the patient and community partner panel, and included access to psychosocial supports and peer support, inclusive of children, families, and caregivers. Black patients’ emotional distress following a breast cancer diagnosis can be further compounded by experiences of medical racism, cultural isolation, and difficulty in accessing psychosocial resources specific to them [60]. Interventions designed specifically for Black women in America have shown significant promise. A recent randomized trial demonstrated that culturally tailored virtual psychoeducational programs improved both physical and emotional quality of life amongst African-American women with breast cancer [61]. Similarly, a recent study highlighted that psycho-oncology interventions, which are culturally grounded, can alleviate distress and improve the ability to cope with the physical and emotional toll of cancer [62]. Psycho-oncological interventions have also been noted to strengthen the trust between patients and providers, which is essential for communities that have been historically marginalized in healthcare [62].

Our findings highlight the desire for greater access to disaggregated data on treatment effectiveness, clinical outcomes, and treatment-related toxicities to facilitate treatment decision-making. While few articles in the literature reported on this need (11.8%; 4/34), despite Canada’s universal, single-payer healthcare system, financial support was identified in the NG and ranked as high priority by our patient and community partner panel. Many of the existing studies examining race-based differences in treatment and outcomes are from the United States using administrative data, as race data are routinely collected in Medicare and Medicaid data holdings [63]; similar variables are not currently systematically collected within Canadian healthcare billing data [64,65,66]. As differences in historical contexts, patterns of breast cancer care, and diversity of the patient population can impact the generalizability of the findings, improved capture of self-reported race and ethnicity variables to facilitate Canadian analyses is warranted. On the policy side, without the availability of data on race or ethnicity, it is difficult to quantify current disparities in access to or receipt of care by patients who are Black, so it is difficult to advocate for improved equity [67]. Current reports by the Public Health Agency of Canada and the Canadian Public Health Association call attention to the matter of Black Canadians experiencing disproportionate burdens of chronic disease, poor access to healthcare, and higher levels of distress due to systemic racism [66,68,69]. In the absence of robust race-disaggregated data, these concerns remain invisible in both policy and practice, reinforcing the urgency of equity-focused research based in Canada. Efforts to address this gap should remain mindful of historic exploitations by research and build in anti-oppressive data sovereignty and governance structures as foundational components.

Our findings must be interpreted within the limitations of our study. A major limitation of our study is generalizability outside of the North American historical and cultural context. Observed differences between the published literature from the United States and our needs assessment of Canadian patients in this study highlight the need for context-specific understanding of patients’ unmet supportive care and informational needs. There have been few articles published to date examining the needs of breast cancer patients who self-identify as Black, and high heterogeneity between the published studies, necessitates the use of a scoping review methodology to narratively synthesize the findings rather than the use of more rigorous meta-analytic methods. Our search excluded articles written in languages other than English due to logistical constraints in screening and abstracting the articles, which may have biased findings through systematic exclusion of articles from outside of North America, Europe, and Australia. Our study samples for both the NG (n = 3) and patient and community partner panel (n = 9) were small. Despite larger populations of Black-identifying patients in many regions of the United States, sample sizes were comparable to previously published works in this space. This reflects the poor engagement of racialized communities in health research [70], resulting from exploitation by researchers and subsequent distrust [71,72]. While our sample included patients of a spectrum of ages from across Canada, participants were largely from Ontario, and the sample did not include participants from either Quebec or Nova Scotia, which have large Black-identifying patient populations. Additionally, our sample was skewed towards those with a middle or higher income, those self-identifying as originally from the Caribbean, and was not inclusive of men, trans, or intersex people who may have unique supportive care and informational needs. While taking an integrative review methodological approach allowed us to compare and validate our findings from the NG and patient and community partner panel against the broader literature [23], further validation of our findings is needed more broadly in the community, inclusive of patients from across the African diaspora and those representative of additional intersecting identities.

## 5. Conclusions

Understanding the needs of Black-identifying patients with breast cancer is imperative in informing culturally tailored interventions to address these needs, and policy and funding structures that support improvement in the delivery of equitable care. Patient navigation was identified as a foundational need, with financial support, access to culturally tailored information and resources, culturally relevant care, racialized data for treatment decision-making, and emotional support identified as high priority. Future work should focus on improving representation and diversity in prospective studies, and on leveraging community-based participatory approaches [73], which centre research activities around patient and community voices to co-create community-driven and culturally tailored supportive care interventions.

## Figures and Tables

**Figure 1 curroncol-32-00580-f001:**
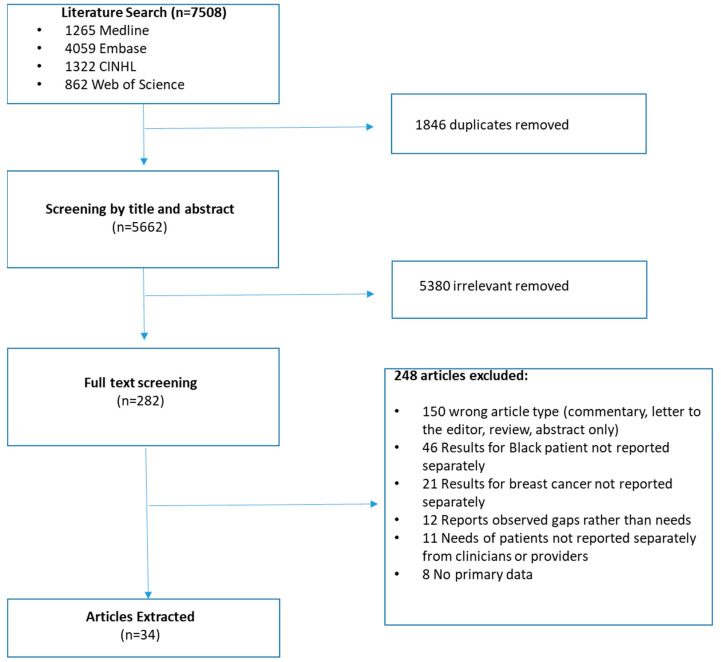
PRISMA diagram.

**Figure 2 curroncol-32-00580-f002:**
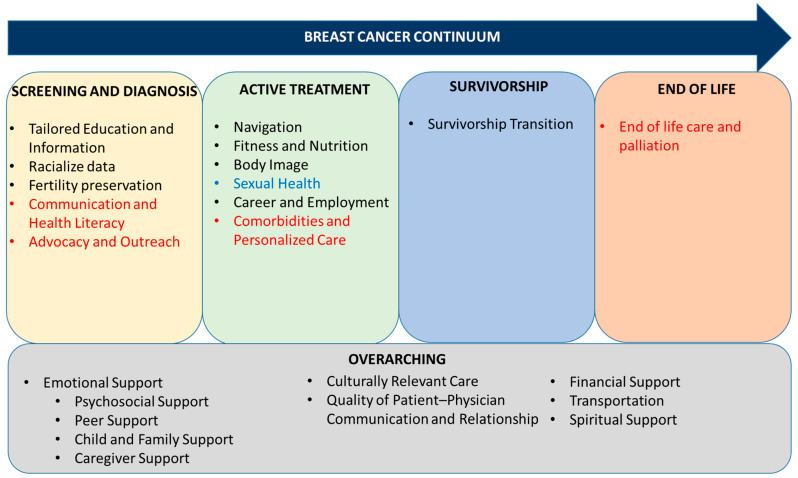
Informational and supportive care needs of Black-identifying patients with breast cancer from both the literature review and NG (black text), the literature review only (blue text), or the NG only (red text), arranged across the breast cancer continuum.

**Figure 3 curroncol-32-00580-f003:**
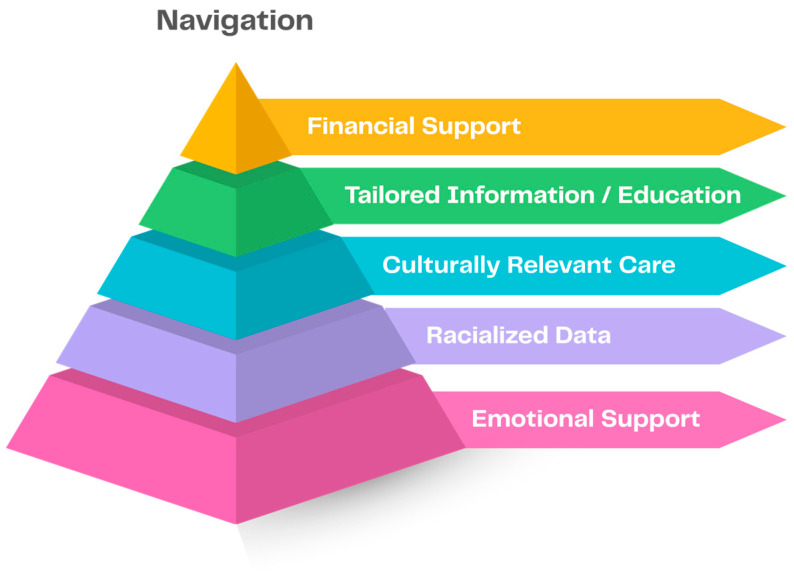
Final list of high priority supportive care and information needs for Canadian patients with breast cancer who self-identify as Black, with navigation as a foundational need.

**Table 1 curroncol-32-00580-t001:** Characteristics of retained articles (n = 34).

Characteristics		N (%)
Year of publication	1997	1 (2.9%)
	2001	3 (8.8%)
	2005	1 (2.9%)
	2006	2 (5.9%)
	2008	1 (2.9%)
	2009	1 (2.9%)
	2010	2 (5.9%)
	2012	1 (2.9%)
	2013	2 (5.9%)
	2014	2 (5.9%)
	2015	1 (2.9%)
	2016	2 (5.9%)
	2017	4 (11.8%)
	2018	3 (8.8%)
	2019	1 (2.9%)
	2020	2 (5.9%)
	2021	2 (5.9%)
Country	United States	34 (100)
Region	South	15 (44.1)
	East Coast	10 (29.4)
	West Coast	3 (8.8%)
	Mid-West	3 (8.8%)
	Multiple	3 (8.8%)
Setting ^1^	Urban	11 (32.4%)
	Rural	1 (2.9%)
	Both	2 (5.9%)
	Not reported	20 (58.8%)
Stage of Cancer Continuum ^1^	Screening and Diagnosis	26 (76.5%)
	Treatment	24 (70.6%)
	Survivorship	8 (23.5%)
	Palliative and End of Life	-
	Overarching	32 (94.1%)

^1^ Categories are not mutually exclusive, and articles could contribute to more than one.

**Table 2 curroncol-32-00580-t002:** Details of each need identified from the literature (n = 34) and/or the NG, with supporting quotes from the NG and the panel.

Stage of the Cancer Continuum; n (%)	Needs; N (%)		Literature Definition	Summary of Nominal Group Discussion	Nominal Group and Panel Quotes
**Overarching;** **32 (94.1%)**	**Emotional Support; 27 (79.4%)**	**Psychosocial Support; 13 (38.2%)**	Care that addresses the psychological and social needs of patients [31]. Emotional support describes listening, confiding, mental health, relationships, social functioning, and being present during the cancer experience [31].	Providers that reflect the Black community Need to connect with resources when you actually need them Clear expectations communicated upfront regarding length of support, wait time to receive support, costs involved with accessing them Limited access within cancer care centres Providers who can be confided in and who will actively listen, to address changing relationships, social needs, and mental health across the cancer journey	“Hospitals need to do better with that emotional and mental support.” “…people need access, especially in our community, to social workers and psychologists and things like that, and it’s tough.”
		**Peer Support; 21 (61.8%)**	Emotional, informational, and practical support given by individuals with shared lived experiences to others facing similar health challenges [31].	Intimidating to enter peer support groups and services that are primarily White Need to be paired with someone who looks like you Someone who has been through the same experience Helpful to hear stories and be able to share Important to have support groups with psychosocial support on-site Important for support groups to have a trained facilitator Need to happen in spaces where participants are comfortable visiting Need to be connected to and embedded within the Black community	“Pair people together that are going through treatment that, you know, have similar backgrounds or have been through it already. Just like you’re there to like, ask questions because doctors are telling me that, like, ‘You’re young. It’s going to be good, like you’re going to get through it.’”
		**Child and Family Supports; 12 (35.3%)**	Support for family and children. This can consist of childcare services, as well as personal support within the home to assist with daily living. It also entails support from family/friends and a desire for longevity to spend time with children/grandchildren and relationships [32].	Resources for childcare to reduce barriers to attending treatments and supportive care appointments Need for tailored family supports to address changes in family dynamics	“…it can really impact family dynamics.”
		**Caregivers Supports; 1 (2.9%)**	Caregiver support encompasses the emotional, informational, and practical resources that enable caregivers to manage their responsibilities, including access to support groups, knowledge sharing, workplace flexibility, transportation assistance, and help with daily tasks [33].	Changes in roles and family dynamics as a result of the breast cancer diagnosis Need to address caregiver burn-out	-
	**Culturally Relevant Care; 5 (14.7%)**		Culturally relevant care is care that is respectful of and aware of sociological differences that are present in the Black community, which can improve health outcomes and patient satisfaction via shared decision-making and improved healthcare delivery [22,34].	“Cancer is not a white thing.” Need care that takes an anti-racist and anti-oppressive lens Prevalence of provider discrimination based on accents Providers treat fewer Black patients and have less knowledge and patients are left feeling like they are outliers or diminished in research Providers historically diminish the pain of Black women Patients are reluctant to seek care because of prior dismissal Need for community outreach programs Need for clinicians with greater cultural awareness and competency, particularly around treatment-related hair loss, skin toxicities, and scarring	“If you have a family doctor, [they could say], ‘hey, Black women’s [risks] are this, so maybe because of that, you should actually be, you know, screened at 40.’ “…healthcare workers need to be sort of encompass [ing] an anti-racist, anti-oppression lens when dealing with racialized communities”
	**Quality of Patient–Physician Communication and Relationship; 12 (35.3%)**		High-quality care is characterized by active listening, culturally sensitive communication, and patient involvement in care decisions [32].	Lack of communication about what supports are available, both in the hospital and in the community Continuity of care is lacking; patients see different providers at every visit, so it is difficult to build rapport/trust Providers lack an understanding of different sociodemographic characteristics and their intersecting impact on healthcare experiences (race, income, geographies) Perpetual self-advocacy is exhausting	“My follow-ups would look like I had no idea. I was like, what happens now? No one was telling me.” “I don’t know what follow-ups are going to look like. So then when I got to meet the new doctor, she’s like, OK, we’re just going to do a yearly follow up on mammogram and like, so I’m going to be getting the same screening as someone that’s in the 40s that hasn’t had breast cancer like, that doesn’t make sense to me. So I had to advocate to get additional screening.”
	**Financial Support; 9 (26.5%)**		Financial support assists patients in covering the direct and indirect costs of medical care, including treatment expenses, transportation, lost income, and other out-of-pocket costs, aiming to reduce financial stress and improve access to care [34].	Need for information on financial planning, critical injury insurance, and financial assistance Need resources to help navigate time off and dealing with employers/employee rights Hardships with working while experiencing treatment-related toxicities or potential loss of income	“So I’ve been part of this Black organization called CAUFP… they connect financial services professionals within the black community.”
	**Transportation; 3 (8.8%)**		Having access to transportation for appointments, screening, treatments, or emergencies [31]. Lack of transportation can be a significant barrier for many patients [31,35].	Need for transportation to and from appointments	“…if you can’t get to your appointments, then you can’t access care.”
	**Spiritual Support; 16 (47.0%)**		Care that is respectful of a person’s beliefs, values, and sense of meaning, especially in the context of illness, suffering, or end-of-life [34,35,36]. There is a belief in divine healing and the importance of faith in the face of bad news, as well as the reliance on faith to maintain social standing [35,36].	Cultural importance of spiritual and religious supports in line with personal beliefs for some patients Access to resources and information tailored to patients’ spiritual beliefs	“So, one of the things I’ve found is that you know there is a big difference in terms of how people get information and sometimes you know whether it’s about literacy or language you know, it’s not one size fits all. There’s also the religious factor here in terms of resources unlike you know, the US where I mean, there are some organizations that you know are specific, like Christian Approach to Cancer Care…”
**Screening and Diagnosis;** **26 (76.5%)**	**Tailored Education and Information; 23 (67.6%)**		Providing breast cancer patients with adequate information is essential for quality care [37]. Satisfaction with information is linked to better emotional, functional, and social well-being, improved coping, and higher treatment adherence. It may also strengthen family communication and a sense of competence in managing illness [37,38].	Need for information on genetic testing Need for information on treatment options and outcomes, specifically amongst Black patients Resources specific to Black hair, including the efficacy of cold capping Where to access prosthetics for darker skin tones Tailored information on skin toxicities, including radiation burns and scarring Need for resources in relevant literacy levels and languages Most existing resources are from the United States; there is a need for more relevant, local resources	“…access to culturally specific supports such as hair… Our hair is different. Our skin is different. So having all those supports that specifically address our unique differences.”
	**Racialized Data; 4 (11.8%)**		Data for women that are disaggregated by race [32]. Racial discordance in data may result in communication barriers, and these barriers often lead to unequal access to health information and inadequate patient participation in healthcare decision-making, which exacerbate racial disparities in health outcomes [32].	Race data are not being collected Lack of inclusion of Black-identifying patients in clinical trials Data are not available for patients to make informed decisions about their treatment and care Need for tailored information on breast cancer risk, treatment effectiveness in Black patient populations, clinical outcomes, and treatment-related toxicities	“…availability of data, that speaks to Black people and that there is a lack of sort of racialized data. I think that was an important point to make.”
	**Fertility Preservation; 3 (8.8%)**		For younger women, cancer treatments may cause premature menopause, infertility, and negative psychosocial effects [39,40]. As such, women considering having children have the option to preserve eggs for future use [39,40].	Access to information on fertility preservation, and resources/assistance with costs Patients need to be given adequate time to make decisions and adequate time to recover, as they usually have a very short period to decide between diagnosis and starting treatment Cultural significance of family and children in Black communities Need emotional support for decision-making, grief, and changing family dynamics Considerations concerning ageism (who is offered information and resources for fertility preservation)	“As most of us know, within the Black community, children and fertility are definitely valued. Perhaps not being able to have kids after treatment, dealing with that and getting support within the community, I think that definitely has cultural relevance.”
	**Communication and Health Literacy—**not in literature review		N/A	Gaps in access to information and education for patients who do not speak English and/or are new immigrants Support to help understand treatments, tests, and results The role that health literacy plays in being able to find and access supports and informational resources, and to self-advocate	“So one of the things I’ve found is that you know there is a big difference in terms of how people get information and sometimes you know it’s about literacy or language. You know, it’s not one size fits all.”
	**Advocacy and Outreach—** not in literature review		N/A	Need for the healthcare system to make more intentional effects to engage with the Black community Need for the cancer system to meet people where they are and engage with them within the community Opportunities for education and to ask questions Need for better connection with and awareness of community-based resources and needs	“…Do you send people who look like them into the community to do the outreach, like where are you advertising? Because it tends to be the case whether it’s organizations or hospitals that they will tend to provide this information to predominantly white women and you know, those things need to change.”
**Active Treatment;** **24 (70.6%)**	**Navigation; 13 (38.2%)**		In navigation, the relationship between patients and oncology providers plays a critical role in supporting treatment adherence. Building trustworthy and long-term connections not only encourages patients to complete active treatment but also fosters ongoing engagement in follow-up care [41,42].	Exhausting to self-advocate during appointments for the care you need Need for a navigator to walk you through what questions to ask, let you know what resources are available and connect you to them, discuss genetic testing, and consider your comorbidities Information on when to present at the hospital if unwell vs. managing at home	“…a physical body to take you through those appointments. And there’s like the navigation of that care.”
	**Fitness and Nutrition; 9 (26.5%)**		Healthy eating and regular physical activity can promote survivorship amongst women with cancer [40].	Access to physiotherapy Access to prehabilitation Access to information on best practices, including how to build stamina to endure treatment Understanding what to eat while on treatment, and how to maximize nutrition/nutritional benefits based on the food that patients already eat Access to cooking resources with recipes that patients are actually likely to eat	“…my oncologist was. ‘Oh, don’t exercise. Don’t do this. Don’t do that.’ And just common sense tells me that. OK, but don’t you want to be in your top physical fitness as much as you can before you have surgery? Because I’m thinking it’s gonna really suck if you’re weak and you have surgery.”
	**Body Image; 7 (20.6%)**		Certain treatments for breast cancer may result in alterations to the body. These changes may impact women’s physical and mental health [40].	Importance of hair, hair texture, body image, and identity Changes in appearance can cause distress and affect quality of life and life enjoyment Need resources around the bereavement of the impending loss of body image and body parts	“I didn’t feel like every morning I had to be reminded about cancer, every single morning by having to colour in my eyebrows. Cause sometimes you don’t want to talk about cancer.”
	**Sexual Health; 6 (17.6%)**		Breast cancer treatments can alter the body and reproductive system. Sexual health addresses at how women involve themselves with intimacy and the opportunity to have children [39].	-	-
	**Career and Employment; 1 (2.9%)**		Being diagnosed with breast cancer hinders one’s career and employment. An indefinite hiatus from employment can negatively impact one’s financial status and ultimately affect their treatment process [35].	Need for information on employee rights, particularly around what you have to tell employers and taking time off work Hardships of working during treatment and while experiencing treatment-related toxicities Potential for loss of income or facing unemployment while undergoing cancer treatment	“She had lost her job… it was super stressful because, you know, you’re alone. Her family was in the Caribbean, and you know, now you have to navigate this financial piece.”
	**Comorbidities and Personalized care—**not in literature review		N/A	Cancer providers are only concerned with cancer-related symptoms and diagnoses, but there is a need for whole-person care Need to understand how to manage the cancer diagnosis alongside other comorbidities such as diabetes, sickle cell disease, and heart disease	“…whether it’s the Black community or Latin American community like you know, some other communities of color, there are also other issues in terms of comorbidity. That like, don’t get addressed. So for example, you know diabetes. That’s not factored in in terms of I guess personalized treatment plans?
**Survivorship;** **8 (23.5%)**	**Survivorship Transitions; 8 (23.5%)**		Throughout this process of moving from treatment to survivorship, there remains a fear of possible remission [31].	Resources within the cancer disappear at the end of treatment Need for greater support during the transition to survivorship care Need more information on what is going to happen during survivorship and what should be happening Fear of cancer recurrence Need for better awareness of community-based resources in this phase and community connection with others who have been through the same transition in care	“…you do feel like you have a lot of support during active treatment. Everyone’s there and then like as soon as you’re done active treatment, it’s like literally; I wasn’t even told. Like what my follow-ups would look like, I had no idea- like I was like what happens now? Like no one was telling me.” “…and then like when I was done treatment, it literally felt like it was like you’re done and like you don’t have access to this anymore. And I was like, whoa, this is like, actually, when I really need this support and it just felt like it was not there.”
**End of Life;** **0 (0%)**	**End-of-Life Care and Palliation**—not in literature review		N/A	Access to palliative care and hospice Access to information and resources for end-of-life support, tailored to individual cultural and religious beliefs Trustworthy information and access to medically assisted dying for those interested in learning about or accessing it	“All of this needs to you know, be in place from start to finish, even in terms of uh diagnosis, not just you know through journey but all the way through end of life with like specific resources for metastatic care and hospice care and for me, my support came from the African American community, whether was like the Young Survival Coalition or other organizations in the US because I have found, you know, a sense of community there and I felt Accepted.”

## Data Availability

The original contributions presented in this study are included in the article/Appendix A. Further inquiries can be directed to the corresponding author(s).

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
