# Peer review of "Supportive Care Needs of Patients with Breast Cancer Who Self-Identify as Black: An Integrative Review"

_curroncol, 2025, doi:10.3390/curroncol32100580_

Round 1
Reviewer 1 Report
Comments and Suggestions for Authors
Thank you for the opportunity to review this valuable integrative review.
The writer identified many strengths of this paper, with one suggestion to add more re: nominal group and literature differences.
Thank you for the opportunity to review this valuable integrative review of psychosocial needs among Black-identifying people with breast cancer.
Brief Summary:
The authors provide an integrative review (scoping review with nominal consensus group of Canadian Black-identifying people with Breast cancer) of current studies of psychosocial needs among Black people with breast cancer. Further, patient partners and community advocacy groups participated in a validation and priority setting exercise.
The Method is new to this writer, but references indicate it has been described for at least 2 decades.
Methods
Strengths:
• Use of established scoping review methods, (PRISMA)
• The integrative review method addressed the scoping review finding of no Canadian studies of Black-identifying patients with breast cancer and added a nominal group (NG) exercise with Black Canadian patients with breast cancer and a validation and priority setting exercise with patient partners and community advocacy groups.
• The participatory action research reduced the usual power hierarchy between researchers and patients through co-design of a supportive care program for Blackidentifying patients with breast cancer. A series of workshops led to consensus discussions to finalize ranking of psychosocial needs.
Results
• Figure 2 & 3 and Table 2 very helpful. The qualitative data and NG & panel quotes are valuable
• NG added nuance and detail e.g. culturally relevant care, fertility preservation – more time, ageism
• NG identified some differences from lit review e.g. spiritual support; sexual health
Discussion is generally thoughtful, extensive and builds on the findings.
Questions for discussion.
• Could the authors comment on differences between lit & NG generally and/or specifically, for example. (might it be NG size which the authors mention as a limitation, methodological e.g. surveys in literature, etc)
o spiritual support in literature (47%) and less pronounced in NG (p. 6, lines 191-193)
o NG identified needs for communication & health literacy and advocacy & outreach p. 7, lines 231-237
o sexual health concerns between the literature and NG?
Author Response
Reviewer 1:
- Could the authors comment on differences between lit & NG generally and/or specifically, for example. (might it be NG size which the authors mention as a limitation, methodological e.g. surveys in literature, etc)
- spiritual support in literature (47%) and less pronounced in NG (p. 6, lines 191-193)
- NG identified needs for communication & health literacy and advocacy & outreach p. 7, lines 231-237
- sexual health concerns between the literature and NG?
The observed differences in the need for spiritual support may reflect both representativeness of the nominal group participants, and differences in the diversity of religious beliefs amongst Black-identifying patients with breast cancer in Canada vs the United States. Identification of communication & health literacy, and advocacy & outreach in the NGs but not the literature is likely reflective of sociodemographic differences in our NG sample relative to previously published articles from the United States which were primarily carried out in the South and/or in jurisdictions with lower median incomes, and lower rates of literacy. Sexual health concerns were identified in the literature but not in the NG, which is likely reflective of the NG size. As per the reviewer’s comment, language has been added to the Results section of the manuscript to address these observed differences.
Reviewer 2 Report
Comments and Suggestions for Authors
The article presents a very relevant topic for improving health equity in Black women with breast cancer. While women experience discrimination in health centers due to their gender, this is even more the case for Black women, which is further exacerbated by the limited number of studies on the subject. For this reason, I congratulate the authors, as they not only conducted a literature review but also explored these barriers in these women, transforming the article into a method for improving the quality of care and living conditions for Black women with this disease.
However, I think the article would gain coherence if these barriers were mentioned or detailed from the introduction, even though they have not been reported in this population or with this disease.
Author Response
Reviewer 2:
- The article presents a very relevant topic for improving health equity in Black women with breast cancer. While women experience discrimination in health centers due to their gender, this is even more the case for Black women, which is further exacerbated by the limited number of studies on the subject. For this reason, I congratulate the authors, as they not only conducted a literature review but also explored these barriers in these women, transforming the article into a method for improving the quality of care and living conditions for Black women with this disease. However, I think the article would gain coherence if these barriers were mentioned or detailed from the introduction, even though they have not been reported in this population or with this disease.
Thank you to the reviewer for their comment. We have updated the Introduction section accordingly to highlight how a lack of representation in prospective trials, a lack of race and ethnicity data in routinely collected administrative health data, and a lack of targeted research are limiting current knowledge around disparities in care and the needs of Black-identifying patients.
Reviewer 3 Report
Comments and Suggestions for Authors
This manuscript is an integrative review examining the supportive care needs of Black patients with breast cancer. The authors synthesize existing literature to identify unmet needs across psychological, social, informational, and financial domains, and discuss systemic barriers contributing to disparities. The review highlights gaps in culturally tailored supportive care interventions and calls for improved equity-oriented research and practice.
Suggestions and concerns:
- The title, “Supportive Care Needs of Patients with Breast Cancer Who Are Black”, should be reformulated. The current wording may sound awkward or potentially insensitive. More standard alternatives would be: “Supportive Care Needs of Black Patients with Breast Cancer” or “Supportive Care Needs of Patients with Breast Cancer Who Self-Identify as Black”. Such phrasing aligns better with current academic style and avoids unintended insensitivity.
- The abstract is clear and informative, but the methodology of the integrative review could be stated more explicitly (databases, timeframe, inclusion criteria). Standard structure (Background, Methods, Results, Conclusions) may be a plus.
- Introduction: Provides useful context about disparities in supportive care among Black patients with breast cancer. Could however benefit from a more explicit framing of why supportive care disparities matter beyond treatment access (e.g., psychosocial, spiritual, and financial domains).
- Methods: The review is integrative, but transparency could be improved - Specify databases searched, time range, and inclusion/exclusion criteria. Clarify whether any assessment of study quality was performed. Without these details, there is a risk of selection bias and omission of relevant evidence.
- Results: Synthesis of included studies is clear and identifies common supportive care gaps.The summary table is useful, but would be stronger if study quality or level of evidence were indicated.
- The generalizability of findings outside the North American continent context should be explicitly acknowledged.
- Discussion: Correctly emphasizes structural and systemic factors contributing to disparities. Could better highlight what is novel compared to prior reviews.
- Limitations should be expanded: narrative synthesis, potential omission of non-English studies, lack of formal quality appraisal.
- Recommendations for culturally tailored supportive care are important but should be more explicitly linked to the evidence reviewed.
- Conclusion: Appropriate but somewhat broad. Should emphasize the importance of culturally tailored interventions and the need for further prospective studies in diverse populations.
Author Response
Reviewer 3:
- The title, “Supportive Care Needs of Patients with Breast Cancer Who Are Black”, should be reformulated. The current wording may sound awkward or potentially insensitive. More standard alternatives would be: “Supportive Care Needs of Black Patients with Breast Cancer” or “Supportive Care Needs of Patients with Breast Cancer Who Self-Identify as Black”. Such phrasing aligns better with current academic style and avoids unintended insensitivity.
Thank you for your comment; we had gone back-and-forth regarding the wording of the title since participants in the nominal groups did self-identify as Black; however, it was unclear from many of the retained articles in the literature review if the participants had self-identified as Black or were racially classified by researchers. In light of the reviewer’s comment, we have updated the manuscript title.
- The abstract is clear and informative, but the methodology of the integrative review could be stated more explicitly (databases, timeframe, inclusion criteria). Standard structure (Background, Methods, Results, Conclusions) may be a plus.
Due to abstract word limits (max 200 words) we had omitted the additional details from the abstract but have added them per the reviewer’s comment. The journal submission guidelines mandate an unstructured abstract so we did not edit the structure of the abstract in order to conform to the style requirements.
- Introduction: Provides useful context about disparities in supportive care among Black patients with breast cancer. Could however benefit from a more explicit framing of why supportive care disparities matter beyond treatment access (e.g., psychosocial, spiritual, and financial domains).
As per the reviewer’s comment, the Introduction has been updated to highlight that, in addition to clinical differences and structural barriers to care, social determinants of health, including anti-Black racism, economic insecurity, and medical mistrust, further exacerbate disparities in the health outcomes of Black women and necessitate additional supportive resources.
- Methods: The review is integrative, but transparency could be improved - Specify databases searched, time range, and inclusion/exclusion criteria. Clarify whether any assessment of study quality was performed. Without these details, there is a risk of selection bias and omission of relevant evidence.
Details of the scoping review methods are included in the second and third paragraphs of the Materials and Methods section of the manuscript. To identify existing studies reporting on the needs of Black-identifying patients with breast cancer, we undertook a scoping literature review. Relevant citations published from database inception to January 2025 were retrieved from Medline, Embase and CINHL using search terms grouped by key concept (breast cancer, support needs and Black), with syntax and headings translated as appropriate across databases (Table S1). Resultant citations were imported into Covidence (Veritas Health Innovation; Melbourne, Australia) for review and duplicates were removed. The review was carried out in accordance with the PRISMA guidelines extension for scoping reviews. Studies were included if they reported on primary studies examining the needs of Black-identifying patients with breast cancer of any gender or sex. The search was restricted to only those articles published in English and reporting on adult populations. Articles were excluded if they were literature reviews, protocols for needs assessments without the accompanying findings, commentaries, letters to the editors or abstract only. Additionally, articles reporting on population-level, cohort studies that derived patient needs based on observed gaps in care, rather than soliciting needs directly from patients were excluded.
- Results: Synthesis of included studies is clear and identifies common supportive care gaps.The summary table is useful, but would be stronger if study quality or level of evidence were indicated.
The literature search was carried out in accordance with the PRISMA guidelines extension for scoping reviews. For scoping style reviews, assessment of quality of the level of evidence is not required since the evidence is assumed to be scant, and insufficient to warrant systematic review and meta-analyses.
- The generalizability of findings outside the North American continent context should be explicitly acknowledged.
Thank you for this comment; generalizability outside of North America has been added to the Discussion section as a limitation of the work. Observed differences between the published needs assessments from the United States and the findings of our Canadian needs assessment highlight the need for context-specific understanding of unmet cancer supportive care and informational needs.
- Discussion: Correctly emphasizes structural and systemic factors contributing to disparities. Could better highlight what is novel compared to prior reviews.
As per the reviewer’s comment we have edited the first paragraph of the Discussion to highlight what is novel about this research. To our knowledge, our study is the first to synthesize the literature around supportive care needs of Black-identifying patients with breast cancer, and the first published needs assessment of this population in the Canadian context.
- Limitations should be expanded: narrative synthesis, potential omission of non-English studies, lack of formal quality appraisal.
Scoping review methodology is appropriate for highlighting knowledge gaps and synthesizing the literature when the research question is broad, the published literature is scarce and heterogeneous, and structured synthesis through meta-analysis is not feasible. As a lack of formal quality appraisal and use of narrative synthesis inherent in scoping review methods, we have added a lack of existing literature necessitating the use of a scoping review as a limitation in the Discussion section. We have also added potential omissions from the exclusion of non-English studies as another limitation.
- Recommendations for culturally tailored supportive care are important but should be more explicitly linked to the evidence reviewed.
We acknowledge this gap in the discussion content that the reviewer had identified. A statement that the needs for navigation, culturally-tailored information and education, and culturally-relevant care were explicitly called our as community priorities through our consensus discussions, has been added to the second paragraph of the Discussion section to highlight what the calls for culturally-tailored supportive care and examples of culturally-tailored supports are rooted in. We feel that this change helps improve interpretability of the Discussion contents.
- Conclusion: Appropriate but somewhat broad. Should emphasize the importance of culturally tailored interventions and the need for further prospective studies in diverse populations.
As per the reviewer’s comment, the Conclusion has been updated to explicitly state the need for culturally-tailored interventions, and greater diversity and representation in future prospective studies.
Round 2
Reviewer 3 Report
Comments and Suggestions for Authors
The authors have addressed all my concerns adequately and I believe the manuscript can be published in the current form
Author Response
The authors have addressed all my concerns adequately and I believe the manuscript can be published in the current form.
We thank the reviewer for their comment.